# Prenatal Diagnosis of Congenital Lymphocytic Choriomeningitis Virus Infection: A Case Report

**DOI:** 10.3390/v14112586

**Published:** 2022-11-21

**Authors:** Fanny Tevaearai, Laureline Moser, Léo Pomar

**Affiliations:** 1Department Woman-Mother-Child, Lausanne University Hospital and University of Lausanne, 1011 Lausanne, Switzerland; 2School of Health Sciences (HESAV), HES-SO University of Applied Sciences and Arts Western Switzerland, 1011 Lausanne, Switzerland

**Keywords:** prenatal diagnosis, ultrasound, MRI, fetus, congenital infection, lymphocytic choriomeningitis, LCMV

## Abstract

Lymphocytic choriomeningitis virus (LCMV) is an emerging neuroteratogen which can infect humans via contact with urine, feces, saliva, or blood of infected rodents. When the infection occurs during pregnancy, there is a risk of transplacental infection with subsequent neurological or visual impairment in the fetus. In this article, we describe a case report of congenital LCMV infection, including fetal imaging, confirmed by positive LCMV IgM in fetal blood and cerebrospinal fluid.

## 1. Introduction

Lymphocytic choriomeningitis virus (LCMV) is an enveloped single-stranded RNA virus, isolated for the first time in 1933 from the cerebrospinal fluid of a deceased woman during the encephalitis epidemic in St. Louis, USA [1].

LCMV is part of the arenavirus family. This virus uses rodents, mostly house mice, as its principal reservoir, and a vertical transmission leads to persistence of the virus across generations. As these animals do not express any immune response, they will develop chronic asymptomatic infections [2]. The dissemination of the virus to humans is hence a zoonosis. Indeed, humans can become infected after contact with infected secretions such as nasal discharge, saliva, milk, semen, urine, or feces [3,4,5,6,7]. Once a person is infected, human-to-human infection is possible. Transmission after organ transplant is rare but has been described [8,9]. LCMV can be found worldwide but seems to be more present in North America and Europe. It can be present all year long, but seems to arise more often during the cold seasons, possibly due to the fact that rodents migrate indoors [10,11]. Contaminated subjects are more likely found among people working with infected animals, such as pet store workers or farmers [10]. Among infected subjects, one-third will remain asymptomatic; one-third will develop nonspecific symptoms such as fever, myalgia, or headache; and the last third will develop more severe symptoms affecting the central nervous system, mostly meningitis and meningoencephalitis [4].

Exact data regarding seroprevalence in immunocompetent humans and in wild mice are unclear. Prevalence in wild mice in the USA is estimated between 3% and 20% [2]. Some studies have shown a seroprevalence in adults varying between 0.3% and 6.8% [2,12,13]. In pregnant women, the seroprevalence could vary between 1.6% and 3.9% according to Argentinian and Croatian studies [12,13].

Maternal–fetal transmission is also possible in humans, as the virus can cross the placental barrier [14]. Its ability to replicate in the human placenta may be enhanced during the first trimester, which is the most vulnerable period for fetuses in the case of transplacental contamination [10]. While infection during early pregnancy can lead to abortion, the surviving embryos will mostly develop with neurological and visual impairments. Different strains of LCMV have been described, some defined as “neurotropic”, others as “viscerotropic”; however, it seems that all strains can replicate in the brain, causing a variety of neurologic signs and symptoms [15,16]. In infected fetuses, LCMV strains also mainly target the central nervous system, specifically the brain and the retina [3,4,10,17,18,19,20]. Signs and symptoms described include microcephaly, periventricular calcification, hydrocephaly, neuronal migration anomalies, pachygyria, porencephalic and periventricular cysts, seizures, neurodevelopmental disability, chorioretinal lacunae, panretinal pigment epithelium atrophy, optic nerve dysplasia or atrophy, and reduced caliber of retinal vessels [5]. Other non-neurological manifestations can be found but are rather rare. These include ascites, small for gestational age, cardiomegaly, skin abnormalities, hydrops, hepatosplenomegaly, pleural and myocardial effusion, thymic hypertrophy, anemia, and thrombocytopenia [13]. 

LCMV is not routinely tested during pregnancy, but maternal symptoms or fetal signs suggestive of infection in the context of rodent exposure should alert practitioners to consider the diagnosis of LCMV infection. Serology by indirect immunofluorescence assay was initially used to confirm the diagnosis [21]. Other methods including enzyme-linked immunosorbent assay (ELISA) and polymerase chain reaction (PCR) showed to be contributive to the diagnosis [10,13].

In this case report, we describe a case of congenital LCMV infection with typical cerebral and extra-cerebral signs in the fetus after maternal contamination in the first trimester of pregnancy.

## 2. Materials and Methods

### 2.1. Patient Consent

We obtained written informed consent from the patient to publish de-identified imaging and data.

### 2.2. Imaging

Fetal ultrasonography was performed using E8 and E10 Voluson machines (General Electric’s Healthcare) with transabdominal (RM6C or RAB4-8) and transvaginal (RIC 5-9) transducers. Fetal MRI was performed on a Siemens 1.5 T Prisma-Fit MR scanner, using 3-dimensional, high-resolution T2-weighted axial, sagittal, and coronal sequences, without fetal sedation.

### 2.3. Sample Collection and Microbiological Investigation

Maternal serum was collected after the diagnosis of fetal abnormalities, at 21 weeks gestation (wg). Microbiological investigations performed on maternal serum included testing for toxoplasmosis, cytomegalovirus (CMV), rubella, parvovirus B19 (PB19), herpes simplex virus (HSV), zika virus, and treponema pallidum (TP) antibodies by immunoassays. Maternal serum was not tested for LCMV because of the unavailability of this sample at the time of diagnosis of congenital LCMV infection. Amniotic fluid was collected by amniocentesis at 22 wg and underwent CMV PCR. Fetal blood was collected during the feticide procedure, at 30 wg. Fetal cerebrospinal fluid (CSF) was collected after expulsion. ELISAs were performed on fetal blood and cerebrospinal fluid to detect LCMV antibodies. IgM and IgG were considered positive for a titer ≥1:10 [19].

### 2.4. Genetic Investigations

A conventional karyotype and an array comparative genomic hybridization (CGH) were performed on the amniotic fluid.

### 2.5. Placental and Fetal Examination

After expulsion, the placenta and the fetus were fixed in 4% buffered formalin. After macroscopic examination, placental and fetal tissues were paraffin-embedded and stained using hematoxylin and eosin. Eight-micrometer sections were used for histology.

## 3. Case Report

We report the case of a 22-year-old primigravida woman, who is a farmer and market worker, without medical history before her pregnancy. The patient experienced a wild mouse bite during her sixth wg and presented six days later to the local emergency unit with multiple symptoms including fever (39.7 °C), asthenia, myalgia, and photophobia. Her symptoms progressively disappeared with symptomatic treatments and the patient revealed to be asymptomatic 10 days after the onset of symptoms. An etiological work-up excluded common infections. After a positive pregnancy test during this consultation, an early ultrasound was offered and confirmed an evolutive intrauterine pregnancy. The patient was then advised to consult her gynecologist for the follow-up of her pregnancy. Her pregnancy was uncomplicated up to 21 wg, including a first trimester ultrasound which did not reveal fetal or placental abnormalities. The morphological scan performed at 21 wg revealed a severe bilateral ventriculomegaly, and this patient was referred to a tertiary center. Serologies on maternal serum at 21 wg were negative for HSV, zika virus, PB19, and TP, and concluded past immunity for toxoplasmosis, CMV, and rubella with stable titers in IgG since 6 wg.

Ultrasound at 22 wg included transvaginal neurosonography, which confirmed severe bilateral ventriculomegaly (occipital ventricular horns at 23 and 17 mm) with periventricular hyperechogenicity, thinning of the cortical mantle, irregular cortical ribbon, dysmorphic nuclei caudate including calcifications, enlarged peri-cerebral spaces, dilated third ventricle, disruption of cavum septi pellucidi, laminated corpus callosum, thick tectum, and cerebellar hypoplasia (mainly affecting one of the cerebellar hemispheres, transverse cerebellar diameter of 20 mm) (Figure 1).

The systolic velocity of the middle cerebral artery was at 40 cm/s, corresponding to 1.4 multiples of the median (mom) for gestational age. Right unilateral microphthalmia was noted and extracerebral examination revealed fetal growth restriction (FGR) with an estimated fetal weight at 450 g, below the third percentile for gestational age [22,23].

Amniocentesis was offered at 22 wg due to fetal abnormalities. The karyotype was 46, XX. The array CHG did not detect microdeletions nor microduplications in the fetal genome, and the CMV PCR in amniotic fluid was negative. The next ultrasounds at 25 and 28 wg found similar cerebral abnormalities, with an increase in hydrocephaly leading to extreme thinning of the brain parenchyma in the right hemisphere and macrocephaly with a head circumference of 292 mm at 28 wg (>97th percentile). The last ultrasound also found a hypoplasia of the optic chiasm particularly affecting the right optic nerve and posterior tract (diameter of the optic chiasm at 5.2 mm, diameter of the right optic nerve and posterior tract at 2.0 and 1.8 mm, Figure 2). Extra-cerebral examination found cardiomegaly, minor pericardial effusion, hepatomegaly, and moderate ascites. The systolic velocity of the middle cerebral artery was 56 cm/s, corresponding to 1.5 mom and suggested fetal anemia.

Fetal MRI performed at 29 wg confirmed ocular and neurological abnormalities and found additional findings: occipital polymicrogyria and subependymal nodules (Figure 3). A multidisciplinary consultation suggested congenital LCMV as the etiology of these fetal infectious signs, and informed the patient of the poor prognosis of the ocular and neurologic lesions, and the different options for managing suspected fetal anemia: fetal blood sampling and in utero transfusion or close fetal monitoring with lung maturation and preterm birth if hydrops develops.

The patient required termination of pregnancy due to poor fetal prognosis. A feticide was performed at 30 wg and included a fetal blood sampling which revealed moderate thrombocytopenia (105,000 platelets/mm3) and anemia (hemoglobin at 8.8 g/dL). A 940 g (<third percentile) female was delivered with a head circumference of 30 cm (>97th percentile). CSF was sampled after delivery, and the fetus and the placenta underwent pathological examinations after formalin fixation. The autopsy of the fetus confirmed extra-cerebral abnormalities (cardiomegaly and pericardial effusion, hepatomegaly, ascites) and found a hypertrophic spleen. Neuropathological examination confirmed hydrocephaly with a rupture of the cavum septi pellucidi leaflets. Examination revealed an irregular cortical ribbon with several areas of fusion of the molecular layers, and the presence of a lamina dissecans suggesting focal polymicrogyria, particularly in the right occipital and temporoparietal lobes (Figure 4). A thin cortex with cell loss and heterotopic neurons has been described. Non-confluent subependymal nodules were confirmed in the right parieto-temporal lobe. The cerebellum showed global dysplasia, with a reduced right hemisphere (white matter loss) and abnormal cerebellar foliation on the left hemisphere. The vermis was hypoplastic and the tectum was enlarged without evidence of a tectal plate tumor. Ocular examination confirmed the presence of unilateral microphthalmia with a hypoplastic optic nerve and found multiple chorioretinal scars on the right eye (Figure 4). Examination of the placenta found non-specific villitis.

LCMV IgM were identified in fetal blood and CSF by ELISA (Table 1).

## 4. Discussion

A total of 70 cases of congenital LCMV were collected in the recent review by Pencole et al. This review includes 11 cases with descriptions of prenatal imaging [13]. The case we described shares some of what appear to be typical neurological signs suggesting LCMV: hydrocephaly, periventricular calcifications, laminated corpus callosum, thinning of the cortical mantle, polymicrogyria, and optic tract hypoplasia. The LCMV exhibits a strong tropism for the brain, mainly for the neuroblasts, which could induce migration and cortical development abnormalities [5]. An inflammatory response is also triggered by LCMV and could explain inflammation-induced lesions of the brain [10,18]. As previously described by Bonthius et al., manifestations of LCMV depend on the host age [18]. This hypothesis was confirmed using a rat model: LCMV infection at the early embryonic stage can disrupt brain development, resulting in cerebral embryopathy [18]. Ocular lesions found in our case include microphthalmy, which seems to occur to a lesser extent in congenital LCMV infections [4]. Other signs observed in this case were FGR, cardiomegaly, minor pericardial effusion, hepatomegaly, and moderate ascites. In the review by Pencole et al., FGR was found in three cases [13]. Various reasons can lead to FGR, including placental and fetal causes. In the case of maternal–fetal contamination, the virus passes the placenta and can cause villitis or intervillitis. This inflammation is responsible for a decrease in placental microvascularization, leading to a reduced nutritional intake and thereby intrauterine growth retardation. Although placental abnormalities have not been described or associated with LCMV, these abnormalities are well known in other maternal–fetal infections [24]. Other interesting signs highlighted in our patient were fetal anemia and moderate thrombocytopenia, also found in three and one patients included in the review by Pencole et al. Fetal anemia related to congenital infections is usually caused by viral affinity for erythroid progenitor cells [25,26], and can induce effusions and hydrops. Fetal thrombocytopenia could reflect a direct cytopathic effect of the virus on megakaryocytes or an indirect immune-mediated effect [27]. Weather LCMV has an increased affinity for erythrocytes or megakaryocytes is unknown, but these mechanisms could be the reasons for fetal anemia and thrombocytopenia observed.

Only two cases of congenital LCMV infection were confirmed during pregnancy by positive PCR in amniotic fluid, described by Delaine et al. in 2018 [28] and Meritet et al. in 2009 [29]. In other cases of congenital infection described, diagnosis was made postnatally by PCR or serology [5,13]. In our case, LCMV infection was diagnosed by the identification of LCMV IgM in fetal blood and CSF. Placental and fetal samples were not tested by real-time reverse transcription PCR or multiplex PCR, as these tests were only performed in specialized laboratories when this case occurred several years ago, and the samples are no longer available for PCR testing, which is one of the main limitations in attributing the observed lesions to LCMV. Cross-reactivity between arenaviruses has been described in immunofluorescence assays, and the specificity of ELISA for confirming congenital LCMV infection from fetal samples is unknown [30]. However, it seems unlikely that other arenaviruses could have caused these fetal abnormalities, as LCMV is the only one to have shown its capacity to cross the placental barrier and to have a worldwide distribution.

Other first-line differential diagnoses are TORCH (Toxoplasma gondii, Rubella virus, CMV, Herpes simplex, and Treponema pallidum) pathogens [3,13]. As mentioned in multiple articles, CMV and toxoplasmosis are probably the most difficult to differentiate from LCMV because all three can be the cause of microcephaly, intracranial calcification, and chorioretinitis. While CMV and toxoplasmosis are more often associated with systemic anomalies, LCMV is only rarely associated with extracerebral signs [2,3,4,10]. Other important differential diagnoses to discuss are Aicardi and Aicardi–Goutières syndromes. Aicardi–Goutières syndrome can have different inheritance patterns: mainly an autosomal recessive mode (caused by inherited mutations or neo-mutations in the ADAR, TREX1, RNASEH2A, RNASEH2B, RNASEH2C, and SAMHD1 genes), and less frequently an autosomal dominant mode (caused by neo-mutations in the IFIH1, TREX1, or ADAR genes). Depending on the affected gene(s), this syndrome can mimic cerebral abnormalities found in LCMV, but ocular manifestations are not described in Aicardi–Goutières syndrome [10,31]. Aicardi syndrome is an X-linked dominant encephalopathy that can mimic ocular signs and cortical abnormalities found in LCMV, but the typical pattern of Aicardi syndrome in prenatal imaging includes corpus callosum agenesis, distortion of the inter-hemispheric fissure, and arachnoid cysts that are not described in LCMV infections [32,33,34]. Other differential diagnosis are the pseudo-TORCH syndromes. These rare genetic disorders also mimic LCMV, but differ as pseudo-TORCH syndromes are often associated with dysmorphism and never associated with chorioretinitis [10] (Table 2).

## 5. Conclusions

Congenital LCMV infection should be kept in mind in cases of exposure to rodents associated with congenital anomalies, all the more if TORCH infections are excluded. As information concerning the virus is growing, more and more cases will most likely be described and diagnosed, allowing better knowledge for the future. Moreover, clinicians should be aware of LCMV, as simple prevention maneuvers can be useful in order to avoid congenital abnormalities related to this infection.

## Figures and Tables

**Figure 1 viruses-14-02586-f001:**
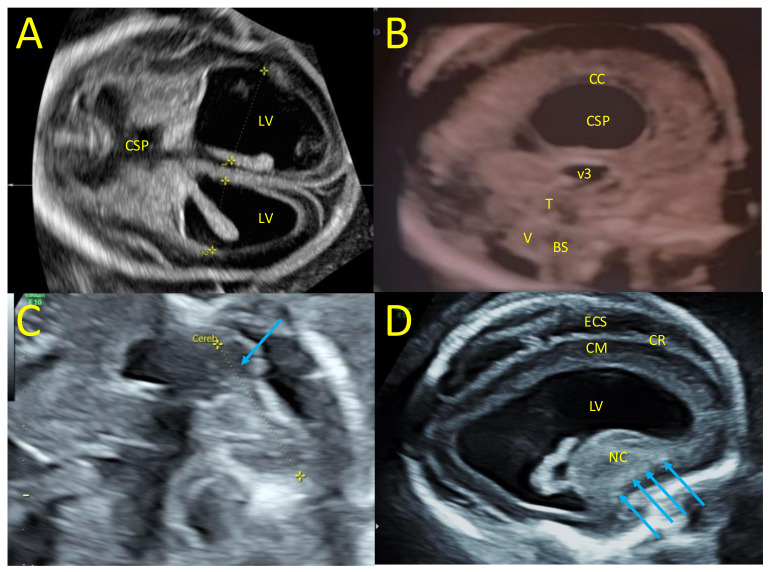
Fetal neurosonography at 22 wg. (**A**) Trans-ventricular axial plane with severe bilateral ventriculomegaly, lateral ventricles (LV) at 23 mm and 17 mm, and disruption of the cavum septi pellucidi (CSP). (**B**) Mid-sagittal plane with laminated corpus callosum (CC), dilated third ventricle (3 V), thick tectum (T), hypoplasia of the vermis (V), and brainstem (BS). (**C**) Trans-cerebellar axial plane with dysplasia of a cerebellar hemisphere (blue arrow). (**D**) Para-sagittal plane with enlarged LV, periventricular hyperechogenicity, thin cortical mantle (CM), irregular cortical ribbon (CR), enlarged peri cerebral spaces, dysmorphic nuclei caudate with calcifications (blue arrows).

**Figure 2 viruses-14-02586-f002:**
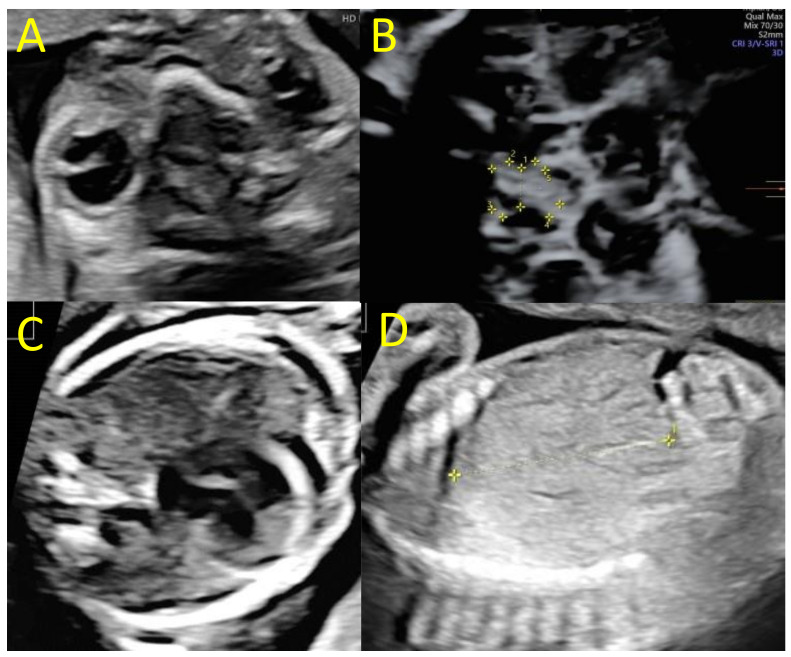
Fetal ultrasounds at 25 and 28 wg. (**A**) Trans-orbital axial plane with unilateral right microphthalmia (25 wg), (**B**) axial plane of the optic chiasm showing a global hypoplasia (1 = diameter of the optic chiasm at 5 mm) affecting the right tracts (2 = diameter of the right optic nerve at 2 mm; 5 = diameter of the right posterior tract at 1.8 mm, 28 wg), (**C**) cardiac four-chamber plane, with cardiomegaly and pericardial effusion (25 wg), (**D**) parasagittal plane of the fetal liver showing a hepatomegaly (diameter of the right liver lobe at 4.3 cm, 25 wg).

**Figure 3 viruses-14-02586-f003:**
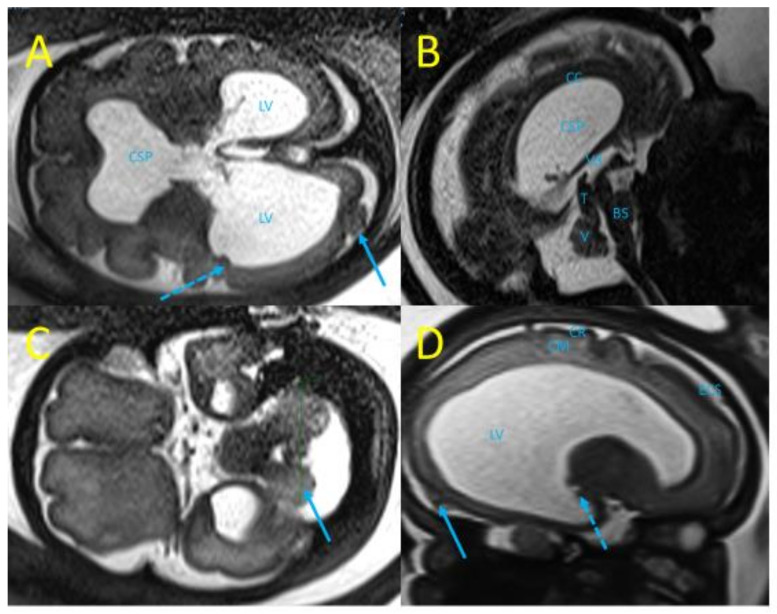
Fetal MRI at 29 wg (T2 weighted sequences). (**A**) Trans-ventricular axial plane with severe bilateral ventriculomegaly, lateral ventricles (LV) at 27 mm and 21 mm, disruption of the cavum septi pellucidi (CSP), subependymal nodules (dot arrow), and occipital polymicrogyria (arrow). (**B**) Mid-sagittal plane with laminated corpus callosum (CC), dilated third ventricle (3 V), thick tectum (T), hypoplasia of the vermis (V), and brainstem (BS). (**C**) Trans-cerebellar axial plane with dysplasia of a cerebellar hemisphere (blue arrow). (**D**) Para-sagittal plane with enlarged LV, subependymal nodules (dot arrow), atrophy of the cortical mantle (CM), irregular cortical ribbon (CR), suggestive of occipital polymicrogyria (arrow).

**Figure 4 viruses-14-02586-f004:**
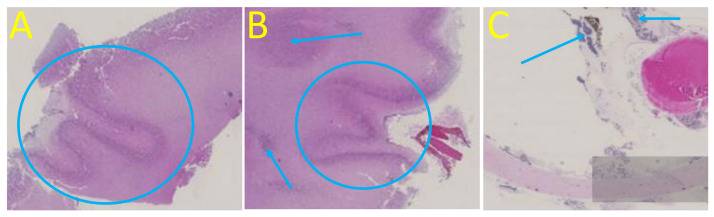
Post-mortem cerebral and ocular examination, hematoxylin and eosin staining, original magnification x10. (**A**) Right occipital cortical plate with polymicrogyria and extreme thinning of the subplate. (**B**) Right temporal polymicrogyria with lamina dissecans, heterotopic neurons (arrow), and periventricular nodule (blue circle). (**C**) Chorioretinal scars (arrows) with hyperpigmentation suggestive of chorioretinitis on the right optic disc.

**Table 1 viruses-14-02586-t001:** Results of the LCMV ELISA performed in fetal samples.

Sample	IgG Titer	IgM Titer
Fetal blood	Positive (1:128)	Positive (1:20)
Cerebrospinal fluid	Negative	Positive (1:16)

IgG and IgM titers were estimated using enzyme-linked immunosorbent assays.

**Table 2 viruses-14-02586-t002:** Differential diagnoses of LCMV infection.

Finding	Retinopathy	Hydrocephaly	Microcephaly	Intracranial Calcifications	Myocarditis/Cardiac Manifestation	Systemic Manifestations	Anemia	Hydrops
LCMV	+++	+++	+++	+++	+	+	+	+
Toxoplasmosis	+++	+	+	++	+	++		+
Parvovirus B19		+			++		++	+
Rubella	+	+	+	+−	+++	+++		+−
CMV	+	+	+++	++	+	+++	+	+
Syphilis	+	+−				++	+	+
Zika	+	+	+++	+++	+	+	+	+−
Aicardi–Goutières		+	+	+++		+		
Aicardi	+++	++	+					

Note: Frequency of findings. +− (rare), + (uncommon), +++ (most common).

## Data Availability

Non applicable.

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
