# Peer review of "Prenatal Diagnosis of Congenital Lymphocytic Choriomeningitis Virus Infection: A Case Report"

_viruses, 2022, doi:10.3390/v14112586_

Round 1

Reviewer 1 Report

The article by Tevaerarai and colleagues reports on an import and clinically often not considered endemic pathogen that can cause devastating cerebral malformations in fetuses. While this report is one of several recent reports on LCMV infections in humans, a report on this pathogen should be included in a special edition on emerging virus infections in pregnancy.

Major points:

1. Based on the information provided by the authors, an autopsy was done. It would be important to provide a more detailed neuropathological description including histological images. In particular a description of the inflammatory changes would be of interest.

2. While detection of immunoglobulins (in particular IgM) against LCMV are a strong indication of infection, it would be important to confirm presence of LCMV via PCR. There are a number of laboratories in Europe that would be able to do this and it should be possible to obtain suitable genetic material from paraffin embedded brain tissue.

3. The discussion on differential diagnosis could be structured more clearly and/or a table included to help the readers appreciate overlap of symptoms / unique symptoms between the diseases listed (infectious and non-infectious). 

Minor points:

1. In the discussion the authors correctly discuss Aicardi-Goutieres syndrome. However, some details are incorrect. For example, inheritance depends on which (of currently 9 identified genes) is affected. 

2. The language needs some revision. Examples: 

2a: line 12: transplacental contamination. I assume the authors mean infection? If so, I would clearly state this. (see also line 28)

2b: line 23: not sure what "ladder" refers to?

2c. line 24-25: please check sentence structure

2d. line 26: consider removing "with"

2e. line 58 consider changing "checked" to "tested for"

2f. line 80: "week's". I don't think it the "'" is correct. 

Author Response

The article by Tevaearai and colleagues reports on an import and clinically often not considered endemic pathogen that can cause devastating cerebral malformations in fetuses. While this report is one of several recent reports on LCMV infections in humans, a report on this pathogen should be included in a special edition on emerging virus infections in pregnancy.

We thank the reviewer for this positive comment and for their help to improve our manuscript.

Major points:

  1. Based on the information provided by the authors, an autopsy was done. It would be important to provide a more detailed neuropathological description including histological images. In particular a description of the inflammatory changes would be of interest.

We thank the reviewer for this suggestion. We have detailed the description of the histo-pathological examination and provided images of the brain and retina:

  • “The autopsy of the fetus confirmed extra-cerebral abnormalities (cardiomegaly and pericardial effusion, hepatomegaly, ascites) and found a hypertrophic spleen. Neuropathological examination confirmed hydrocephaly with a rupture of the cavum septi pellucidi leaflets. Examination revealed an irregular cortical ribbon with several areas of fusion of the molecular layers, and the presence of a lamina dissecans suggesting focal polymicrogyria, particularly in the right occipital and temporoparietal lobes (Figure 4). A thin cortex with cell loss and heterotopic neurons has been described. Non-confluent subependymal nodules were confirmed in the right parieto-temporal lobe. The cerebellum showed global dysplasia, with a reduced right hemisphere (white matter loss) and abnormal cerebellar foliation on the left hemisphere. The vermis was hypoplastic and the tectum was enlarged without evidence of a tectal plate tumor. Ocular examination confirmed the presence of unilateral microphthalmia with a hypoplastic optic nerve and found multiple chorioretinal scars on the right eye (Figure 4). Examination of the placenta found non-specific villitis.

Figure 4: Post-portem cerebral and ocular examination A) Right occipital cortical plate with polymicrogyria and extreme thinning of the subplate. B) Right temporal polymicrogyria with lamina dissecans, heterotopic neurons (arrow) and periventricular nodule (dot arrow). C) Chorioretinal scars (arrows) with hyperpigmentation suggestive of chorioretinitis on the right optic disc.”

  1. While detection of immunoglobulins (in particular IgM) against LCMV are a strong indication of infection, it would be important to confirm presence of LCMV via PCR. There are a number of laboratories in Europe that would be able to do this and it should be possible to obtain suitable genetic material from paraffin embedded brain tissue.

The case we report occurred six years ago, and we no longer have material available to test by PCR except for brain sections that were fixed with formalin prior to autopsy. To our knowledge, formalin fixation prevents RNA virus detection by PCR. A reference laboratory had already been contacted prior to publication and had indicated that LCMV detection by RT-PCR on formalin-fixed sections was not possible in their institution. However, if the reviewer could indicate us a laboratory where brain sections could be sent to perform PCR testing, we would be happy to do it.

We fully agree that a positive PCR could confirm the causality of LCMV on the lesions observed, and we propose to add this limitation in the discussion:

  • “In our case, LCMV infection was diagnosed by the identification of LCMV IgM in fetal blood and CSF. Placental and fetal samples were not tested by real-time reverse transcription PCR or multiplex PCR as these tests were only performed in specialised laboratories when this case occurred several years ago, and the samples are no longer available for PCR testing, which is one of the main limitations in attributing the observed lesions to LCMV. Cross-reactivity between arenaviruses has been described in immunofluorescence assays, and the specificity of ELISA for confirming congenital LCMV infection from fetal samples is unknown[30]. However, it seems unlikely that other arenaviruses could have caused these fetal abnormalities, as LCMV is the only one to have shown its capacity to cross the placental barrier and to have a worldwide distribution.”

  1. The discussion on differential diagnosis could be structured more clearly and/or a table included to help the readers appreciate overlap of symptoms / unique symptoms between the diseases listed (infectious and non-infectious).

We have added a table including mentioned differential diagnosis:

“Table 2. Differential diagnoses of LCMV infection

Finding

Retinopathy

Hydrocephaly

Microcephaly

Intracranial calcifications

Myocarditis/Cardiac manifestation

Systemic manifestations

Anemia

Hydrops

LCMV

+++

+++

+++

+++

+

+

+

+

Toxoplasmosis

+++

+

+

++

+

++

+

Parvovirus B19

+

++

++

+

Rubella

+

+

+

+-

+++

+++

+-

CMV

+

+

+++

++

+

+++

+

+

Syphilis

+

+-

++

+

+

Zika

+

+

+++

+++

+

+

+

+-

Aicardi-Goutieres

+

+

+++

+

Aicardi

+++

++

+

Note: Frequency of findings. + (rare), + (uncommon) to +++ (most common).”

Minor points:

  1. In the discussion the authors correctly discuss Aicardi-Goutieres syndrome. However, some details are incorrect. For example, inheritance depends on which (of currently 9 identified genes) is affected.

We have modified the discussion to read:

  • “Aicardi-Goutières syndrome can have different inheritance patterns: mainly an autosomal recessive mode (caused by inherited mutations or neo-mutations in the ADAR, TREX1, RNASEH2A, RNASEH2B, RNASEH2C, and SAMHD1 genes), and less frequently an autosomal dominant mode (caused by neo-mutations in the IFIH1, TREX1 or ADAR genes).”

  1. The language needs some revision. Examples:

The corrections have been made.

2a: line 12: transplacental contamination. I assume the authors mean infection? If so, I would clearly state this. (see also line 28)

2b: line 23: not sure what "ladder" refers to?

2c. line 24-25: please check sentence structure

2d. line 26: consider removing "with"

2e. line 58 consider changing "checked" to "tested for"

2f. line 80: "week's". I don't think it the "'" is correct.

Reviewer 2 Report

The case report is interesting, well documented and well discussed; 

Gynecologist must be aware that LCMV can induce severe fetal damages. 

Author Response

We thank the reviewer for their very positive comment. We fully agree that perinatal healthcare providers should be aware that LCMV can induce CNS and ocular abnormalities in the fetus. 

Reviewer 3 Report

In this case-report, an interesting case of congenital LCMV infection with typical cerebral and extra-cerebral signs in the fetus was described. The patient related a wild mouse bite during her sixth wg with multiple symptoms including fever (39.7°C), asthenia, myalgia, and photophobia. Though she eventually got better 10 days after symptoms onsets, a severe bilateral ventriculomegaly for fetal at 21 wg was revealed by.trimester ultrasound. And the LCMV IgM were identified in fetal blood and CSF by ELISA . This case presented well on the whole, and emphased that congenital LCMV infection should be kept in mind in cases of exposure to rodents associated with congenital anomalities when TORCH infections are excluded. There are some issues need to be noticed as listed below:

1.For sentence in lines 31-33, references are needed here.

2.In line 48 and lines 182-184, whether all strains of LCMV target mostly the central nervous system?.

Author Response

Reviewer’s comments #3:

In this case-report, an interesting case of congenital LCMV infection with typical cerebral and extra-cerebral signs in the fetus was described. The patient related a wild mouse bite during her sixth wg with multiple symptoms including fever (39.7°C), asthenia, myalgia, and photophobia. Though she eventually got better 10 days after symptoms onsets, a severe bilateral ventriculomegaly for fetal at 21 wg was revealed by.trimester ultrasound. And the LCMV IgM were identified in fetal blood and CSF by ELISA . This case presented well on the whole, and emphased that congenital LCMV infection should be kept in mind in cases of exposure to rodents associated with congenital anomalities when TORCH infections are excluded. There are some issues need to be noticed as listed below:

We thank the reviewer for their positive comment. The answers to the questions and changes in the text can be seen below.

1.For sentence in lines 31-33, references are needed here.

The references have been added.

2.In line 48 and lines 182-184, whether all strains of LCMV target mostly the central nervous system?

We thank the reviewer for this interesting comment. In fact, there are many different LCMV strains which are separated in two categories: ‘neurotropic’ and ‘viscerotropic’ stains. Although it appears that these different strains can all replicate in the brain, ‘viscerotropic’ viruses could favorizing viral hepatitis, enterocolitis, and myocarditis.

We have modified the introduction to include this comment:

  • “Different strains of LCMV have been described. Some defined as ‘neurotropic’, others as ‘viscerotropic’, however it seems that all strains can replicate in the brain, causing a variety of neurologic signs and symptoms [15,16]. In infected fetuses, LCMV strains also mainly target the central nervous system, specifically the brain and the retina [3,4,10,17–20]. Signs and symptoms described include microcephaly, periventricular calcification, hydrocephaly, neuronal migration anomalies, pachygyria, porencephalic cysts and periventricular cysts, seizures, neurodevelopmental disability, chorioretinal lacunae, panretinal pigment epithelium atrophy, optic nerve dysplasia or atrophy, and reduced caliber of retinal vessels [5]. Other non-neurological manifestations can be found but are rather rare. These include ascites, small for gestational age, cardiomegaly, skin abnormalities, hydrops, hepatosplenomegaly, pleural and myocardial effusion, thymic hypertrophy, anemia and thrombocytopenia [13]”

Round 2

Reviewer 1 Report

The revised manuscript reads very well. However, the methods section needs updating to include the histology; also, a size bar is required for the histological images.

Author Response

We thank the reviewer for their help to improve this case report. 

The manuscript has been revised to include the histology in the methods and to include a size bar for the histological images.